# Cheminformatics Identification of Phenolics as Modulators of Penicillin-Binding Protein 2a of *Staphylococcus aureus*: A Structure–Activity-Relationship-Based Study

**DOI:** 10.3390/pharmaceutics14091818

**Published:** 2022-08-29

**Authors:** Jamiu Olaseni Aribisala, Saheed Sabiu

**Affiliations:** Department of Biotechnology and Food Science, Faculty of Applied Sciences, Durban University of Technology, P.O. Box 1334, Durban 4000, South Africa

**Keywords:** phenolics, penicillin-binding protein, structure–activity-based virtual screening, allosteric, molecular dynamic simulation

## Abstract

The acquisition of penicillin-binding protein (PBP) 2a in resistant strains of *Staphylococcus aureus* allows for the continuous production of cell walls even after the inactivation of intrinsic PBPs. Thus, the discovery of novel therapeutics with enhanced modulatory activity on PBP2a is crucial, and plant secondary metabolites, such as phenolics, have found relevance in this regard. In this study, using computational techniques, phenolics were screened against the active site of PBP2a, and the ability of the lead phenolics to modulate PBP2a’s active and allosteric sites was studied. The top-five phenolics (leads) identified through structure–activity-based screening, pharmacokinetics and synthetic feasibility evaluations were subjected to molecular dynamics simulations. Except for propan-2-one at the active site, the leads had a higher binding free energy at both the active and allosteric sites of PBP2a than amoxicillin. The leads, while promoting the thermodynamic stability of PBP2a, showed a more promising affinity at the allosteric site than the active site, with silicristin (−25.61 kcal/mol) and epicatechin gallate (−47.65 kcal/mol) having the best affinity at the active and allosteric sites, respectively. Interestingly, the modulation of Tyr446, the active site gatekeeper residue in PBP2a, was noted to correlate with the affinity of the leads at the allosteric site. Overall, these observations point to the leads’ ability to inhibit PBP2a, either directly or through allosteric modulation with conventional drugs. Further confirmatory in vitro studies on the leads are underway.

## 1. Introduction

*Staphylococcus aureus* is a Gram-positive bacterium that has remained a major public health concern over the years, due to its resistance to major antibacterial therapies [1]. For instance, Methicillin-resistant *S. aureus* (MRSA) is currently one of the most common global causes of infective endocarditis, as well as skin and respiratory system infections [2,3]. In many countries, strict policies are in place regarding the use of antibiotics, to keep the organism vulnerable due to the limited treatment options that are available [1,4]. However, even with limited usage, the number of strains of MRSA with reduced sensitivity or outright resistance to currently available antibiotics, such as daptomycin, ceftaroline, vancomycin, amoxicillin, linezolid, and cefotaxime have risen in recent times [2,4,5,6,7]. Thus, in addition to raising public awareness about safe antibiotic usage, developing new therapeutics with novel modes of action to circumvent and keep up with the increased rate of antibiotic resistance remains a top priority for many antimicrobial drug development research groups.

The survival of bacteria is dependent on the integrity of their cell walls, which are made up of peptidoglycan as the primary building unit [8,9]. Peptidoglycan consists of a repeating unit of a disaccharide (N-acetyl glucosamine (NAG)–N-acetylmuramic acid (NAM)) with the peptide stems on the NAM unit, and through crosslinking of peptide stems of nearby peptidoglycan, a mature cell wall is formed [10,11]. The peptide stem differs in various organisms and in *S. aureus*, it is a five-chain peptide L-Ala-γ-D-Glu-L-Lys(Gly)_5_-D-Ala-D-Ala with the lysine side chain being linked to pentaglycyl during cell wall crosslinking [10]. The transglycosylase (synthesis of peptidoglycan backbone) and transpeptidase (crosslinking of peptidoglycan) activities during cell wall production are carried out by the penicillin-binding proteins (PBPs) [9]. Due to the essential role of PBPs in cell wall synthesis, which is critical for bacterial survival, they act as an excellent target for antibiotics, particularly the β-lactams [6,10].

*Staphylococcus aureus* possesses four inherent PBPs (PBP1, PBP2, PBP3, and PBP4) [11]. However, due to the acquisition of the *mecA* gene, a fifth PBP known as PBP2a is frequently found in MRSA, conferring antibiotic resistance in the organism [12]. As a result, MRSA can continue cell wall synthesis while other PBPs are inhibited by β-lactam antibiotics [12,13]. Penicillin-binding protein 2a from *S. aureus* is a high-molecular-weight class B PBP identified solely in antibiotic-resistant strains of *S. aureus*. Penicillin-binding protein 2a’s transpeptidase active site, like that of other PBPs, contains the active site serine (S403) at the N-terminus of the SXXK sequence motif [12,14,15]. Interestingly, the activity of this protein is controlled by allosterism at a position, 60 Å away from the active site, where cell wall crosslinking occurs [12,16]. As a result, previously inert β-lactam antibiotics may now reach and inactivate this crucial protein, thereby inhibiting the organism [12,16]. Allosteric modulation is mostly used to treat eukaryotic disorders rather than prokaryotic infections [12]. However, recent advances in the infectiveness of biochemical and computational screening approaches, alongside protein structure characterization, point to the prospects of allosteric activity regulation as a novel concept for antibiotic discovery [12,13,14,16]. Specifically, by exploiting allosteric modulation of PBP2a of MRSA, promising compounds with the capacity to exhibit direct antibacterial activity while also working in synergism with existing conventional antibiotics have been identified [1,16]. Hence, in this study, the active site of PBP2a was employed in screening compounds against MRSA, while the modulation of the active site was studied via the binding of compounds to the PBP2a allosteric site.

Most currently accessible antibiotics were produced from microbial sources, while plant-based antibiotics have been largely ignored, owing to their weaker activity relative to their microbial-derived counterparts [16]. However, with the increased interest in plant-based antimicrobials in recent years, most especially the phenolic compounds, numerous candidates demonstrating promising pharmacological properties have been identified which can be further developed into therapeutics [16,17]. In bacteria, the efficiency of β-lactam antibiotics has been attributed to their ability to penetrate the outer cell membrane, resist inactivation by plasmid- or chromosomal-encoded β-lactamase, and bind efficiently with mutated penicillin-binding proteins (e.g., PBP2a, a target that has been implicated in the resistant strain of *S. aureus*) [9]. These characteristics have been reported in phytonutrients, including phenolics [16,17]. Phenolics’ ability to effectively bind and modulate bacterial druggable targets, such as β-lactamase, and consequently inhibit various multidrug-resistant Gram-negative and Gram-positive bacteria has been reported [16,17], while also enhancing cell membrane permeability for antimicrobial absorption [18]. In addition, being the most frequently occurring metabolite in plants, phenolics offer other health and therapeutic benefits as antioxidants and against degenerative diseases, such as cancer and diabetes [16,17,19,20,21]. These characteristics of phenolics point to their prospective antibacterial capabilities for treatment or synergistic effects with conventional antibiotics in the treatment of infections caused by multidrug-resistant *S. aureus*. Thus, with over 10,000 phenolics isolated over the years and reported in the literature with different pharmacological characteristics [22], we employed for the first time in this study, structure–activity-relationship-based pharmacophore and molecular docking approaches to screen phenolics with capabilities to inhibit the activities of PBP2a of *S. aureus*. The most promising phenolics were further studied for their thermodynamic compatibility with PBP2a and their ability to modulate the active site of PBP2a following binding at the allosteric site (Figure 1). This was undertaken to identify prospective antibacterial candidates that can treat or act in synergy with conventional antibiotics in the treatment of infections caused by MRSA.

## 2. Materials and Methods

### 2.1. Druggable Target Acquisition, Preparation, and Identification of Binding Sites

The X-ray crystal structure of PBP2a of *S. aureus* (3ZFZ) was obtained from the protein data bank (PDB) (https://www.rcsb.org, accessed on 21 August 2021). The structure was prepared via removal of water molecules and nonstandard amino acids in preparation for molecular docking using the UCSF Chimera v1.15 software tool [23]. The x-y-z coordinates and amino acid residues at the active (centre (X: 28.9; Y: 29.43; Z: 87.60); radius (14.9)] and allosteric sites (centre (X: 11.34; Y: 33.75; Z: 22.74); radius (13.5)] of PBP2a were defined as previously reported using Discovery Studio version 21.1.0 [24] and afterwards validated by literature [16].

### 2.2. Structure-Based Pharmacophore Screening of Phenolic Compounds 

With over 10,000 phenolic compounds currently accessible in the literature [22], a preliminary computational screening was performed, where a library of phenolics against PBP2a was built by utilizing the ZINCPharmer database (http://zincpharmer.csb.pitt.edu, accessed on 21 August 2021). For the screening, the 3D structure of PBP2a from *S. aureus* and a consensus phenolic pharmacophore (derived from 32 phenolics) produced from PharmaGist (https://bioinfo3d.cs.tau.ac.il/PharmaGist/php.php, accessed on 17 August 2021) were used [25]. Characteristics, such as the presence of an aromatic ring, the creation of hydrogen bonds, the charge of the ligand, and the hydrophobic contact between the ligand and the receptor atoms, were utilised to identify 1550 phenolics that interacted with the active site of PBP2a in *S. aureus* (Appendix A).

### 2.3. Ligand Retrieval, Optimization, and Molecular Docking at the Active and Allosteric Site of PBP2a 

The docking strategy was used to screen down the constructed phenolics library at the active site of PBP2a using Python Prescription (PyRx) v 0.9.5. (Figure 2). Before docking, the library of phenolics and reference β-lactam antibiotics (amoxicillin, cefotaxime, aztreonam, and doripenem) [16] were optimised through the addition of Gasteiger charges using the Open Babel program plug-in on PyRx [24]. Following that, docking of the prepared protein (PBP2a) and optimised ligands (made up of the built library of phenolics and reference β-lactam antibiotics) was performed using the AutoDock tool present on PyRx [22]. Docking at the active site of PBP2a was assured by selecting amino acid residues at the active site whose grid box coordinates match the established x-y-z coordinates. Following the selection of the top-twenty phenolics with the highest binding affinity at the active sites of PBP2a using the reference β-lactam antibiotics as the benchmark, compound similarity, pharmacokinetics friendliness, and synthetic feasibility evaluation were employed in selecting the top five from the top-twenty phenolics. Afterward, the docking of the top-five phenolics was done at the allosteric site of PBP2a (Figure 3). Here, the top-five phenolics were individually retrieved from PubChem, prepared via charge addition using the UCSF Chimera v 1.15 software program, and then docked at the allosteric site of PBP2a via amino acid selections at the allosteric site (grid box coordinate coinciding with established x-y-z coordinates) using the AutoDock tool present on PyRx. The docked complexes of the top-five phenolics with the most-energy-minimised conformation (maximum binding affinity) at both the active and allosteric sites were afterward retrieved in PDB format for further molecular dynamic (MD) simulation.

However, as molecular docking methods often produce pseudo-positive binding conformations as the most-energy-minimised pose, validation of docking studies is often required. One of the most common ways to evaluate the correctness of docking geometry is to measure the root-mean-square deviation (RMSD) of the ligand from its reference position in the answer complex after optimal superimposition [26]. A low RMSD value of <1 between the docked ligand from its reference position in the answer complex suggests the same binding orientation, which encouraged docking technique validation [27]. In this study, validation of the docking pose was performed via the superimposition approach against the experimental co-crystal structure of PBP2a from *S. aureus* (3ZFZ). The superimposition showed that the top-five phenolics and amoxicillin achieved the same orientation with the native inhibitor of 3ZFZ with a low RMSD value of <1, which validated the docking scores observed in the study; these are presented in Figure 2 and Figure 3 for the PBP2a active and allosteric sites, respectively.

### 2.4. Top-Twenty Phenolics Pharmacokinetic Properties Prediction 

The SwissADME web (http://swissadme.ch/index.php, accessed on 15 October 2021) and Molinspiration (https://www.molinspiration.com/cgi-bin/properties, accessed on 15 October 2021) toolkits were used in this study for a robust prediction of hit phenolics’ physicochemical properties, pharmacokinetics, drug-likeness, and medicinal chemistry friendliness, while the toxicological profiles were evaluated using the Protox II webserver (https://tox-new.charite.de/protox_II/, accessed on 15 October 2021). Taken together with the binding affinity of the top-twenty phenolic compounds, the results of these investigations were utilised to narrow down the list to five compounds.

### 2.5. Molecular Fingerprinting of the Top-Twenty Phenolics

Galaxy Europe (https://usegalaxy.eu./#, accessed on 15 July 2022) was used to molecularly fingerprint the top-twenty compounds. In a nutshell, Galaxy Europe’s “molecule to fingerprint” tool was used to convert the compounds’ smile format into “Open Babel FP2 fingerprints.” The “Open Babel FP2 fingerprints” were then clustered using the fingerprinting algorithms “Taylor-Butina” and “NxN clustering”, with thresholds of 0.8 and 0.0, respectively [22].

### 2.6. Molecular Dynamic (MD) Simulations of Top-Five Hit Phenolics

The MD simulation was performed as previously described [28]. To summarise, the AMBER 18 package was utilised, and the simulation was run for 120 ns using the FF18SB variation of the AMBER force field to characterise the operating systems. Similarly, the ANTECHAMBER was employed to generate the atomic partial charges of the ligands by applying general amber force field (GAFF) measurements and constrained electrostatic potential (RESP). The Leap module’s hydrogen atoms, Na+ and Cl− counter ions were used to neutralise the systems. The amino acid residues in each case were suitably numbered, and the systems were suspended inside an orthorhombic box of TIP3P water molecules in such a way that all atoms were within 8 Å of any box edge. The SHAKE method was used to limit the hydrogen atom bonds in each simulated system. Each simulation had a step-size of 2 fs, which corresponded to the isobaric–isothermal ensemble (NPT) with randomised seeding, a temperature of 300 K, a constant pressure of 1 bar, and a Langevin thermostat with a collision frequency of 1.0 ps and a pressure-coupling constant of 2 ps. Following that, the 120 ns MDS findings were reviewed and classified as post-dynamic data.

### 2.7. Post-Dynamic Analysis

The post-dynamic analysis was performed by saving the systems’ coordinates and trajectories throughout the simulation period and then evaluating them post-simulation using the PTRAJ module of the AMBER 18 package. Exploiting the CPPTRAJ module of the same package, the analysis of root-mean-square deviation (RMSD), root-mean-square fluctuation (RMSF), the radius of gyration (ROG), and solvent-accessible surface area (SASA) were carried out and their plots were generated using Origin v 6.0 [29]. Likewise, using the Molecular Mechanics/GB Surface Area (MMGBSA) approach, the binding free energy was estimated using an average among 100,000 snapshots taken from a 120 ns MD simulation trajectory using the equation ΔG_bind_ = G_complex_ − (G_Receptor_ + G_ligand_). The PBP2a-ligand plots, which indicate molecular interactions between ligands and amino acid residues at the PBP2a active site or allosteric site, were identified from the final complexes obtained and visualised with Discovery Studio v 21.1.0 (Dassault Systemes BIOVIA, San Diego, CA, USA) [30].

## 3. Results and Discussion

### 3.1. Ranking of Phenolics against the Active Site of PBP2a of S. aureus

Screening using the structure-based pharmacophore as employed in this study allowed for hit-searching using distinct characteristics specific for phenolics and PBP2a [25]. Using this method, over 10,000 phenolics currently accessible from the literature were narrowed down to 1550 phenolics with capabilities to interact with the active site of PBP2a of *S. aureus* (Appendix A). Following molecular docking, the top-twenty phenolics were identified from the 1550 phenolics. Molecular docking is a structure–activity-based virtual screening technique that enables compounds to be ranked based on their orientations and interactions at the binding site of a protein, and the higher the negative value, the better the affinity of the compound for the protein [28]. In this study, the top-twenty phenolics had higher negative docking scores (−6.8 to −8.5) than all the conventional antibiotics (amoxicillin (−6.2), cefotaxime (−6.0), aztreonam (−5.5), and doripenem (−5.8)) with ZINC03978446 (Isotheaflavin) having the highest negative docking score (Figure 4). This finding regarding the top-twenty compounds could be an indication of their better potential affinity for the PBP2a active site relative to the standards, which also lent credence to the effectiveness of the structure–activity-based pharmacophore strategy employed in this study. This observation agrees with the report of Koes and Camacho [25], who demonstrated the specificity and efficiency of structure–activity-based pharmacophores in screening hits for a target.

Due to the high failure rate at the preclinical and clinical phases of drug development, the pharmacokinetics, drug-likeness, synthetic feasibility, and toxicity characteristics of the top-twenty phenolics were evaluated in ranking the top-five phenolics. Since the evaluation of the pharmacokinetics and medicinal chemistry of compounds via the in silico tools used in this study can sometimes produce pseudo-positive results, different online tools [SwissADME web and Molinspiration toolkits] equipped with models for robust prediction were employed in the study to enable a comparison and validation of the pharmacokinetics and medicinal chemistry information of the phenolics. This was performed in order to have a high confidence level against false-positive results prior to further in vitro and in vivo validations. Interestingly, all the top-five phenolics (ZINC34953149 (silicristin), ZINC71621503 (propan-2-one), ZINC38337968 (epigallocatechin 4-benzylthioether), ZINC95486076 (Chroman-4-one), and ZINC03978503 (epicatechin gallate)) passed the Lipinski rule of five (Figure 4 and Table 1). Lipinski et al. [31] predicted that compounds with less than two violations of the rule of five, which limit the molecular weight to <500 g/mol, the hydrogen acceptor to ≤10, the hydrogen bond donor to ≤5, and the octanol coefficient to <5, will be orally bioavailable. The top-five compounds fulfilling this rule show their ability to be orally bioavailable to reach target sites and exert their pharmacological effects. The top-five phenolics were soluble in water, with a bioavailability score of 0.55 each, similar to that of amoxicillin. Except for ZINC95486076 (Chroman-4-one), with a higher gastrointestinal tract absorption rate, the other top-rated phenolics had similar gastrointestinal tract absorption traits similar to amoxicillin (Table 1 and Figure 4). All these observations are a pointer to the drug-likeness properties of the top-five phenolics. Cytochrome P450 (CYP), being an important isoenzyme in drug metabolism, plays a key role in drug toxicity [32]. Only ZINC03978503 (epicatechin gallate) and amoxicillin were observed to be non-inhibitors of all the CYPP450 isoenzymes (CYP1A2, CYP2C19, CYP2C9, CYP2D6, and CYP3A4), which predict their low capability to cause drug–drug interactions when used with other drugs, as CYPP450 isoenzymes are responsible for metabolizing about 80% of all medications used [32]. ZINC34953149 (silicristin), ZINC38337968 (epigallocatechin 4-benzylthioether), ZINC71621503 (propan-2-one), and ZINC95486076 (Chroman-4-one) all inhibit CYP3A4, which proposes their likely capability to cause drug interaction with other drugs being metabolised by this enzyme. Except for ZINC34953149 (silicristin) and ZINC95486076 (Chroman-4-one), which were predicted to be active for immunotoxicity, the other top-ranked phenolics and amoxicillin were predicted to be inactive for carcinogenicity, hepatotoxicity, mutagenicity, cytotoxicity, and immunotoxicity (Table 1). Moreover, the top-five phenolics belonged to drug toxicity classes (five and four) that are suitable for drug development [6] with less than five synthetic feasibility scores, a limit for compounds that will be less complex to synthesise [33], which hints that structural alterations can be made on them for druggability improvement. Findings from this aspect of the study suggest the medicinal and pharmacokinetics friendliness as well as the drug-likeness properties of the top-five phenolics with their capability to be derivatised and synthesised for improved druggability and reduced toxicity.

### 3.2. Thermodynamic Binding Free Energy of Top-Five Phenolics at the Active Site of PBP2a of S. aureus

The binding free energy measures the difference in energy between a complex (ligand and receptor) and its unbound receptor component and the higher the negative value, the better the ligand’s affinity for the protein [34]. In this study, except for propan-2-one, the top-ranked phenolics had higher binding free energy for the active site of PBP2a of *S. aureus* than amoxicillin. Silicristin, epigallocatechin 4-benzylthioether, chroman-4-one, and epicatechin gallate, when complexed with PBP2a at the active site had −25.61 kcal/mol, −24.75 kcal/mol, −22.34 kcal/mol, and −23.11 kcal/mol, respectively, which were all higher than the −21.54 kcal/mol observed with amoxicillin, with silicristin having the highest binding free energy score (Table 2). This observation is suggestive of the better potential of silicristin, epigallocatechin 4-benzylthioether, chroman-4-one, and epicatechin gallate as inhibitors of PBP2a, with silicristin being the most promising among the top-five phenolics. The top-five phenolics, particularly silicristin, showing better potential for the PBP2a active site than amoxicillin after the 120 ns MD simulation corroborated the molecular docking findings of this study, in which the top-five phenolics all had higher docking scores than amoxicillin, with silicristin having the highest value as well. Generally, findings from the thermodynamic binding free energy investigations of this study revealed the advantage of the top-five phenolics over amoxicillin in the management/treatment of infections caused by MRSA.

### 3.3. Thermodynamic Stability, Compactness and Flexibility of the Top-Five Phenolics at the Active Site of PBP2a of S. aureus

The thermodynamic stability of the top-five phenolics at the active site of PBP2a of *S. aureus* was studied by evaluating the RMSD, ROG, SASA and intramolecular numbers of hydrogen bonds and distances over a 120 ns simulation time. The RMSD trajectory depicts the time-dependent departure of a complex’s structure from its unbound structure and the lower the RMSD value, the closer the complex structure to the unbound structure which depicts stability [34]. According to Ramirex and Caballero [35], an RMSD value of less than 3 Å is generally acceptable and considered good. In this study, during the first 10 ns of the simulation, all the systems (unbound PBP2a and complexed PBP2a) were observed to be equilibrating, a situation in which atoms of the macromolecules (PBP2a) and the surrounding solvent relaxed before the system reached a stationary state [36]. After equilibration, each of the systems was observed to take a more diverse trajectory, an observation which impacted the average RMSD observed after the 120 ns simulation period (Figure 5 and Table 3). More specifically, at the active site, the unbound PBP2a and PBP2a + propan-2-one complex deviated by 4.5 Å and 2.5 Å, respectively, immediately after equilibrating and maintained such deviation with little swaying throughout the simulation (Figure 5). This observation impacted the high average RMSD observed for unbound PBP2a and PBP2a + propan-2-one complex throughout the remaining simulation periods, at 6.86 Å and 5.78 Å, respectively (Table 3). These values are not within the good deviation limit of less than 3 Å and hence suggest that the systems may not enjoy reasonable degrees of stability. On the other hand, the PBP2a + silicristin complex maintained a more stable deviation within the 2 Å and 3 Å range before 60 ns, however after this period, the PBP2a + silicristin complex showed a greater deviation to 10 Å from 3 Å and maintained such deviation throughout the simulation, which also contributed to the high average RMSD at 5.65 Å which may also indicate less stability (Figure 5 and Table 3). Other systems, such as PBP2a + amoxicillin, PBP2a + epigallocatechin 4-benzylthioether, PBP2a + chroman-4-one, and PBP2a + epicatechin gallate all fluctuated within a more stable deviation between 2 and 5 Å during the entire simulation time and had lower average RMSD values of 4.07 Å, 3.49 Å, 3.26 Å, and 3.42 Å, respectively, with PBP2a + chroman-4-one having the lowest RMSD (Figure 5 and Table 3). Although, the RMSD values of the simulated compounds were greater than 3 Å, when compared to the unbound PBP2a system with an RMSD value of 6.86 Å, it could be generally inferred that the binding of PBP2a with the top-five phenolics as well as amoxicillin caused an improved thermodynamic structural stability, and hence indicated their possible prospects as a PBP2a inhibitor. Relative to amoxicillin + PBP2a complex; PBP2a + epigallocatechin 4-benzylthioether, PBP2a + chroman-4-one, and PBP2a + epicatechin gallate all had lower RMSD values which insinuated their advantage over amoxicillin as PBP2a inhibitors and this observation agreed with the results of the thermodynamic binding free energy investigations in this study, where epigallocatechin 4-benzylthioether, chroman-4-one, and epicatechin gallate all had a higher binding free energy than amoxicillin. This finding is consistent with the work of Alhadrami et al. [16], where flavonoids enhanced the stability of *S. aureus* PBP2a when bound at its active site relative to the reference β-lactams investigated. Notably, silicristin having a higher RMSD value than amoxicillin when complexed with PBP2a, suggestive of silicristin’s lesser thermodynamic stability with PBP2a did not support the higher binding free energy observed for silicristin relative to amoxicillin in this study, however, this observation could be attributed to the sudden drop in stability (the increase in RMSD) after 60ns as observed from the RMSD plot (Figure 5) and hence highlights the importance of a longer duration of simulation in drug discovery.

The ROG is another thermodynamic stability metric which measures the time-dependent compactness of a complex during an MD simulation and the lesser the value the more compact and stable the complex [16,29]. In this study, lesser fluctuations in compactness were observed in PBP2a + amoxicillin, PBP2a + epigallocatechin 4-benzylthioether, PBP2a + chroman-4-one, and PBP2a + epicatechin gallate complexes throughout the 120 ns simulation period (Figure 6). This observation could mean that the binding of PBP2a with amoxicillin, epigallocatechin 4-benzylthioether, chroman-4-one, and epicatechin gallate had little impact in causing the folding of PBP2a and this observation was evidenced by their respective average ROG values (37.57 Å, 36.92 Å, 37.84 Å, and 37.19) which were higher than 35.37 Å and 35.34 Å observed for the unbound PBP2a and PBP2a + propan-2-one that enhanced PBP2a folding just after 10 ns of the simulation (Table 3). The binding of silicristin to PBP2a, on the other hand, caused no folding of PBP2a until after 60 ns (Figure 6). Generally, the observations from the ROG findings in this study were inversely proportional to that observed in the RMSD study, which indicated that the higher thermodynamic stability observed in the binding of PBP2a following binding of the top-five phenolics as well as amoxicillin relative to unbound PBP2a is not directly due to the ability of the top-five phenolics and amoxicillin to cause PBP2a folding. This observation is not in accordance with the study by Aribisala et al. [29] where ROG findings of type 2A topoisomerases following binding with metabolites from *C. cujete* correlated with the RMSD findings. However, a previous study by Baig et al. [37] reported that ROG is the mass-weighted root-mean-square distance of a collection of atoms from their common centre of mass, which indicates that the position of binding of ligands relative to the centre of the mass may have impacts on ROG values. In this study, the point of interaction (active site) of PBP2a is far from the centre of the mass and hence may have resulted in the little impact noted in the folding of PBP2a following binding with the top-five phenolics relative to the unbound PBP2a. Furthermore, characteristics, such as protein topology, size, length, point of interactions, and amino-acid packing have also been shown to have impacts on protein folding [38], and could be the likely reasons for the observations with ROG patterns in this study.

Another important thermodynamic stability metric that analyses protein folding and changes in surface area during a simulation is the SASA, with higher SASA values suggesting an increase in protein volume [39]. Similar to the ROG findings, a lesser fluctuation in SASA in PBP2a + amoxicillin, PBP2a + epigallocatechin 4-benzylthioether, PBP2a + chroman-4-one, and PBP2a + epicatechin gallate complexes plots was observed throughout the 120 ns simulation period which points to the lesser impact of amoxicillin, epigallocatechin 4-benzylthioether, chroman-4-one, and epicatechin gallate in reducing the surface area of PBP2a (Figure 7 and Table 3). However, unbound PBP2a and PBP2a + propan-2-one displayed a slight reduction in SASA value, which suggests a reduction in protein volume just after 10 ns and maintained such reduction with minor swaying throughout the simulation period. These observations impacted their overall average SASA values, as the unbound PBP2a (26786.05 Å) and PBP2a + propan-2-one (26,303.34 Å) both had lesser SASA values than the other investigated systems (Table 3 and Figure 7). On the other hand, the binding of silicristin on PBP2a revealed a more stable complex till 60 ns before instigating a reduction in SASA value, which specifies protein folding. This observation further corroborates the RMSD and ROG findings of this study on the importance of a longer duration of simulations in determining the compatibility of a ligand–protein complex. Generally, the observations from the SASA findings of this study were in agreement with the ROG findings that the higher stability observed in the binding of PBP2a with the top-five phenolics as well as amoxicillin relative to unbound PBP2a may not be due to their ability to cause PBP2a folding.

Intramolecular hydrogen bonds and distance are important in the stability of a protein structure and hence can be assessed to understand the impact of ligand binding on the stability of a protein during simulation [34,40]. Figure 8a displayed a stable fluctuation in the pattern of the number of hydrogen bonds formed in PBP2a generally between 300 and 380 before and following bindings of the top-five phenolics and amoxicillin. This observation signifies the non-disruption of the thermodynamic entropy of PBP2a following bindings with the top-five phenolics and amoxicillin. Averagely, the top-five phenolics and amoxicillin, when complexed with PBP2a, resulted in more intramolecular hydrogen bonds (between 341 and 346) than unbound PBP2a at 339 (Table 3 and Figure 8a). This observation suggests the occupancy of some intramolecular space by the ligands and the increase in the intramolecular hydrogen bonds suggests the hydrogen bond interactions between the top-five phenolics as well as amoxicillin with PBP2a. This observation correlates with the RMSD findings of this study, which suggest the thermodynamic compatibility of the top-five phenolics and amoxicillin with PBP2a. The top-five phenolics binding to PBP2a resulted in more intramolecular hydrogen bonds than PBP2a + amoxicillin (341), with PBP2a + silicristin (346) having the highest intramolecular hydrogen bonds. This observation signifies that the complexes formed by the top-five phenolics and most especially by silicristin, formed more hydrogen bonds with PBP2a during the 120 ns simulation time which could have contributed to the higher binding free energy observed with epigallocatechin 4-benzylthioether, chroman-4-one, epicatechin gallate, and silicristin towards PBP2a relative to amoxicillin. Figure 8b displayed a continued reduction in intramolecular-hydrogen-bond distance as the simulation progressed, which was similar in all the systems (unbound-PBP2a- and phenolics-PBP2a-complexed systems). All the systems had an average intramolecular-hydrogen-bond distance of 2.85 Å (Table 3), which suggests that the binding of PBP2a with the top-five phenolics and amoxicillin did not disrupt the arrangement and the original geometry of PBP2a but rather caused more internal pull between the atoms and residues of PBP2a as the simulation progressed. This further corroborates the thermodynamic compatibility of the top-five phenolics and amoxicillin with PBP2a observed in this study as established by the RMSD findings.

The thermodynamic flexibility of amino acid residues following the binding of the top-five phenolics at the active site of PBP2a of *S. aureus* was studied through the measurement of the RMSF value. The RMSF value takes into consideration the average fluctuation of atoms and residues of a protein structure over a simulation time which can be related to their ability to form intra- and intermolecular stable bonds and the lesser the fluctuation, especially at the active site where ligand binding and catalysis takes place, the stronger the bonds and the affinity of the ligand for the protein [39]. In this study, higher fluctuation in residues between 1 and 100, 125 and 300, and 575 and 590, as well as a lesser fluctuation between residues 300 and 575, and 590 and 645 were observed in all the systems with the level of fluctuation varying from one system to the other, which consequently impacted the average RMSF observed (Figure 9a and Table 3). Generally, binding of the top-five phenolics (epigallocatechin 4-benzylthioether (2.27 Å), chroman-4-one (2.49 Å), epicatechin gallate (2.38 Å) propan-2-one (2.45 Å), and silicristin (2.26 Å)) with PBP2a resulted in a lesser average RMSF than the unbound PBP2a (2.71 Å) and PBP2a + amoxicillin (2.85 Å). This observation suggests the lesser flexibility of PBP2a amino acid residues following binding of the top-five phenolics, with silicristin binding of PBP2a being the most prominent, revealing their stronger attraction and ability to promote PBP2a amino acid residue stability. These findings are consistent with the observed thermodynamic binding free energy, where silicristin had the most favourable affinity towards PBP2a. Interestingly, when the active site gatekeeper residue (Tyr446), which has been demonstrated to obstruct drug access to PBP2a [12,16], was further evaluated to understand its flexibility during the simulation, findings showed that Tyr446 had a reduced average RMSF compared with the average RMSF of the whole PBP2a in all the systems (Table 4 and Figure 9b). This observation was more prominent in the bound systems than the unbound system, suggestive of the active involvement of Tyr446 in the binding of the top-five phenolics and amoxicillin. Specifically, except for the PBP2a-epigallocatechin 4-benzylthioether complex (1.76 Å), the binding of PBP2a with the top-five phenolics (propan-2-one (1.67 Å), chroman-4-one (1.53 Å), epicatechin gallate (1.50 Å), and silicristin (1.49 Å)) resulted in a lesser Tyr446 fluctuation than PBP2a + amoxicillin (1.72 Å) and unbound PBP2a (1.92 Å), with PBP2a + silicristin having the lowest value (Table 4, Figure 9b). This observation further determines the affinity of the top-five phenolics and most especially silicristin, with the active site gatekeeper residue being suggestive of their advantage over amoxicillin as PBP2a inhibitors.

### 3.4. Molecular Docking of Top-Five Phenolics at the Allosteric Site of PBP2a of S. aureus

The molecular docking at the allosteric site distal to the active site is based on the fact that when properly occupied, it simultaneously opens the gatekeeper amino acid residue (Tyr446) inside the active site and realigns the conformation of key residues to allow catalysis [41]. Docking at the allosteric site revealed that the top-five phenolics all had higher docking scores than amoxicillin (Table 5). PBP2a + Silicristin, PBP2a + propan-2-one, PBP2a + epigallocatechin 4-benzylthioether, PBP2a + chroman-4-one, and PBP2a + epicatechin gallate had docking scores of −8.4 kcal/mol, −8.3 kcal/mol, −8.1 kcal/mol, −8.0 kcal/mol, and −8.5 kcal/mol, respectively, which were all higher than the −7.7 kcal/mol observed for the PBP2a-amoxicillin complex. This observation could be suggestive of the greater potential affinity of the compounds for the allosteric site than amoxicillin. When compared with the observations at the main active site, the top-five phenolics showed better affinities for the allosteric site (8.0–8.5 kcal/mol) than the active site (6.8–7.5 kcal/mol) (Table 1 and Table 5). However, since molecular docking is only a preliminary indication of how a compound interacts with a protein, further rigorous binding calculation at the allosteric site was carried out using the Molecular Mechanics/GB Surface Area (MMGBSA) approach over a 120 ns simulation period. In addition, in order to understand how simulation at the allosteric site affects the active site, the binding free energies of the top-five phenolics at the allosteric site were correlated with the fluctuation of the active site gatekeeper residue (Tyr446), and studied via RMSF evaluation.

### 3.5. Thermodynamic Binding Free Energy Following 120 ns Simulation of the Top-Five Phenolics at the Allosteric Site of PBP2a of S. aureus

The binding of the top-five phenolics (Silicristin (−33.57 kcal/mol), propan-2-one (−28.55 kcal/mol), epigallocatechin 4-benzylthioether (−36.83 kcal/mol), chroman-4-one (−29.52 kcal/mol), and epicatechin gallate (−47.65 kcal/mol)) with PBP2a resulted in a higher binding free energy than the amoxicillin (−14 kcal/mol) complex at the allosteric site. This observation, while corroborating the molecular docking scores at the allosteric site, further demonstrates the potential affinity of the top-five phenolics for the allosteric site, revealing their probable capability to modulate the amino acid residues of PBP2a allosteric site. Interestingly, this finding is in correlation with the observation of Alhadrami et al. [16] where some flavonoids showed a better affinity at the allosteric site of PBP2a of *S. aureus* than conventional β-lactam antibiotics (e.g., ampicillin and ceftaroline). Interestingly, epicatechin gallate and other top-five phenolics which demonstrated their potential affinity for the allosteric site of PBP2a in the order of epigallocatechin 4-benzylthioether > silicristin > chroman-4-one > propan-2-one, all had a higher binding free energy at the allosteric site than at the active site. This observation demonstrates the better potential affinity of the top-five phenolics and most especially epicatechin gallate for the allosteric site, revealing their potential benefit in treating *S. aureus* infections in combination with standard β-lactam antibiotics.

### 3.6. Allosteric Modulation of PBP2a Active Site Amino Acid Residues Following 120 ns Simulation at the Allosteric Site

The allosteric modulation of the active site gatekeeper residue of PBP2a was studied by evaluating the RMSF of PBP2a through a 120 ns simulation time. Except for epigallocatechin 4-benzylthioether, binding at the allosteric site of PBP2a by the top-five phenolics (silicristin (2.45 Å), propan-2-one (2.49 Å), chroman-4-one (2.55 Å), and epicatechin gallate (2.06 Å)) and amoxicillin (2.56 Å) resulted in a lesser average RMSF than the unbound PBP2a (2.71 Å), with epicatechin gallate having the lowest RMSF value (Table 5). This observation shows the potential of the top-five phenolics and most especially epicatechin gallate in promoting PBP2a internal amino acid stability following binding at the allosteric site. However, further insight into how allosteric binding influences access to the PBP2a active site was achieved by evaluating fluctuations of the gatekeeper residue, Tyr446, which has been demonstrated to hinder drugs from entering the PBP2a active site. The average RMSF of Tyr446 is presented in Table 6 while its plots are presented in Figure 10b. Interestingly, PBP2a allosteric binding of epicatechin gallate (3.61 Å), silicristin (2.91 Å), epigallocatechin 4-benzylthioether (3.12 Å), and propan-2-one (2.27 Å) all resulted in the higher RMSF of Tyr446 relative to the unbound PBP2a (1.66 Å) and PBP2a + amoxicillin (1.30 Å). This observation is suggestive of the higher fluctuation of the gatekeeper residue during the 120 ns allosteric binding of PBP2a with epicatechin gallate, silicristin, epigallocatechin 4-benzylthioether, and propan-2-one. Another inference from this observation is the lesser involvement of Tyr446 in the intramolecular interactions of adjacent residues (α9 helix (Q577–Y588) and M641 residues) of PBP2a thereby resulting in its higher instability in bond formation and its reduced obstruction of the PBP2a active site. This inference correlates with the report of Mahasenan et al. [41], who experimentally demonstrated the shifting of adjacent residues that may form intramolecular interactions with Tyr446 following the allosteric binding of PBP2a with cefepime. Remarkably, the higher fluctuation of the gatekeeper residue following allosteric binding of the top-five phenolics relative to amoxicillin was in agreement with the observed binding free energy findings in this study, where the top-five phenolics demonstrated higher binding free energy than amoxicillin (−14.36 kcal/mol). Specifically, the higher fluctuation of Tyr446 was noted to partially correlate with the binding free energy obtained in this study with the top-five phenolics at the allosteric site of PBP2a (epicatechin gallate (ΔG_bind_: −47.65 kcal/mol; RMSF: 3.61 Å) > epigallocatechin 4-benzylthioether (ΔG_bind_: −36.83 kcal/mol, RMSF: 3.12 Å), silicristin (ΔG_bind_: −33.57 kcal/mol; RMSF: 2.91) > propan-2-one (ΔG_bind_: −28.55 kcal/mol, RMSF: 2.27 Å) > Chroman-4-one (ΔG_bind_: −29.52 kcal/mol, RMSF: 1.63 Å) > amoxicillin (ΔG_bind_: −14.36, RMSF:1.30 Å)) and the graft is presented in Figure 11. This observation pinpoints that the higher the affinity of a compound to the allosteric site of PBP2a, the higher the instability of Tyr446. This finding is also in agreement with the report of Mahasenan et al. [41] who demonstrated both experimentally and computationally how appropriate allosteric site binding of PBP2a simultaneously opened the gatekeeper residue of the active site. Furthermore, the top-five phenolics, when bound to the allosteric site, have higher fluctuations of the gatekeeper residue relative to when they are complexed directly at the PBP2a active site. This observation could be investigative of the gatekeeper residue’s lesser involvement in intramolecular interactions with other residues of PBP2a when the top-five phenolics were bound at the allosteric site rather than intermolecular interactions with the top-five phenolics when bound at the active site. Interestingly, a lesser fluctuation of the gatekeeper residue was noted in amoxicillin when bound at the allosteric site relative to the active site, suggesting that the gatekeeper residue was more actively involved in intramolecular interactions with other PBP2a residues when amoxicillin was bound at the allosteric site compared with intermolecular interactions with amoxicillin when bound at the active site. This observation may be analytical of the advantages of the top-five phenolics in the allosteric modulation of PBP2a over amoxicillin. Generally, findings from this aspect of the study partially correlates the affinity of the top-five phenolics at the allosteric site of PBP2a with the fluctuation of the active site gatekeeper residue (Tyr446). In addition, the study demonstrated the lesser involvement of Tyr446 in the intramolecular interactions with other PBP2a residues when the top-five phenolics bind at the allosteric site compared with their intermolecular interactions at the active site, suggestive of the reduced obstruction of the active site, an observation that was not observed with amoxicillin. Interestingly, binding of PBP2a at the allosteric site caused the enhanced thermodynamic stability of PBP2a as the bound system (4.06 Å) had a lesser average RMSD value than the unbound PBP2a (6.86 Å), with chroman-4-one having the lowest RMSD value (3.24 Å) (Appendix A and Appendix A). Similarly, the ROG and SASA findings showed the orderliness of PBP2a geometry following binding of the top-five phenolics and amoxicillin at the allosteric site (Appendix A and Appendix A). All of these observations suggest the thermodynamic compatibility of the top-five phenolics and amoxicillin at the allosteric site of PBP2a.

### 3.7. Bonds Analysis of the Interaction Plots of the Top-Five Phenolics against the Active and Allosteric Sites of PBP2a of S. aureus

In general, the ability of a ligand to bind and inactivate a protein is determined by a variety of thermodynamic criteria relating to the flexibility of amino acid residues, protein stability and compactness, and, most importantly, the nature of interactions with the essential amino acids of a protein [42,43]. Thus, in this study, the number, nature, and length of bond interactions formed by the top-five phenolics with PBP2a at both the active and allosteric sites were analysed. The plot of interactions for amoxicillin and phenolics with the highest binding free energy at both the active and allosteric sites are presented in Figure 12a,b, respectively, while the plots for other top-five phenolics are presented in Appendix A. The nature, bond lengths, and numbers of interactions of the top-five phenolics as well as amoxicillin at the active site of PBP2a differs from one compound to another and were noted to have impacts on the binding free energy observed in this study. These bond interactions include hydrogen bonds (conventional and carbon), Van der Waals, amide π-stacked, π-anion, π-sigma, π-cation, π-π t-cation, π-π t-shaped, π-π t-stacked, π-alkyl, alky, π- Sulphur, as well as some unfavourable donor–donor interactions. In this study, propan-2-one and amoxicillin (Figure 12a) had the highest number, with 17 interactions each (Table 7 and Appendix A). This observation is not directly in accordance with the free binding energy scores of this study where silicristin, epigallocatechin 4-benzylthioether, chroman-4-one, and epicatechin gallate had a higher free binding energy than amoxicillin and propan-2-one. However, when the interactions were further analysed as to the nature and distance of the bonds formed with amoxicillin and propan-2-one, more insight into why they have a low binding free energy relative to the other compounds was revealed. Hydrogen bond interactions have been demonstrated as one of the most important non-covalent bonds in drug discovery as they exhibit unusually strong intermolecular interactions [40]. Propan-2-one had only one hydrogen bond interaction with Tyr447 (Table 7) relative to four hydrogen bonds each formed by silicristin (Tyr446, Thyr600, Asn464, Glu602) (Figure 12b), epigallocatechin 4-benzylthioether (Lys584, Glu460, Thr582, Asp586), and amoxicillin (Ala642, Ser643, Ser461, Ser462) as well as five hydrogen bonds formed by epicatechin gallate (Lys581, Ser461, Glu447 (2), Gly599) against the active site of PBP2a of *S. aureus* (Table 7 and Appendix A). Although, chroman-4-one also had one hydrogen bond interaction with Ser642 of PBP2a, however, when the average distance of interactions of PBP2a + chroma-4-one (4.99 Å) was compared with PBP2a + propan-2-one (5.53 Å), the average bond length of the former was considerably lower, which could have contributed to its observed higher binding free energy relative to PBP2a + propan-2-one. The importance of bond length observed in this study agrees with the report of Du et al. [34], who demonstrated the importance of shorter bond length in the stronger pull between two intra- or intermolecular entities, which may lead to their atoms being held together more tightly and hence a greater affinity. Similarly, the lower binding free energy of amoxicillin relative to the other top-four phenolics observed in this study could also be partly attributed to its higher bond length, which is second only to silicristin against PBP2a. Furthermore, relative to the top-five phenolics with no unfavourable bonds (except silicristin), amoxicillin had two unfavourable donor-donor bonds which are relatively shorter in length (2.66 Å) when compared to the average bond interaction distance of PBP2a + amoxicillin (5.14 Å). The formation of the unfavourable bonds which imply the presence of a repelling force in the PBP2a + amoxicillin complex [44], and the short distance of the unfavourable bond which suggests how strong the repulsion was, could have greatly impacted the lower binding free energy observed with the PBP2a-amoxicillin complex relative to the top-five phenolics despite amoxicillin forming 17 interactions (of which four were hydrogen bonds) with PBP2a (Table 7 and Appendix A). Although one interaction in the PBP2a-silicristin interaction plot was also unfavourable, with a longer bond length of 6.35 Å relative to the average bond interaction length of PBP2a-silicristin (5.75 Å) (Figure 12b), this implied its lesser impact on the binding free energy observed in silicristin against PBP2a. While the PBP2a-silicristin complex (16) had one interaction greater than the interactions observed with PBP2a-epigallocatechin 4-benzylthioether, PBP2a-epicatechin gallate, and PBP2a-chroman-4-one complexes, having a remarkably higher average bond length did not entirely justify the PBP2a-silicristin complex having a higher binding free energy against PBP2a than the other three complexes. However, among the top-five phenolics as well as amoxicillin, only silicristin interacted with a hydrogen bond with Tyr446, an important amino acid residue at the active site of PBP2a [16,41] (Figure 12b), a clue of its affinity for the PBP2a active site which could have impacted its higher binding free energy observed in this study. Amoxicillin interacted with a π-cation, propan-2-one interacted with a π-π t-shaped, epigallocatechin 4-benzylthioether interacted with a π-Sulphur, while epicatechin gallate and chroman-4-one both interacted with Van der Waals interactions, all of which are of lesser strength relative to the hydrogen bonds. Generally, none of the top-five phenolics as well as amoxicillin interacted with the catalytic residue Ser403 [16] that was sandwiched between the α2-α3 loop and ß3–ß4 loop of the PBP2a active site, suggesting the partial occupancy of the active site by the top-five phenolics and amoxicillin.

Similar to the active site, the nature, bond lengths and numbers of interactions of the top-five phenolics as well as amoxicillin at the allosteric site of PBP2a differs from one compound to another and were noted to have impacts on the binding free energy observed in this study. These bond interactions include hydrogen bonds (conventional, carbon, and π-donor), Van der Waals, amide π-stacked, π-anion, π-sigma, π-cation, π-π t-cation, π-π t-shaped, π-π t-stacked, π-alkyl, alky as well as some unfavourable donor–donor interactions (Appendix A). The PBP2a + epigallocatechin 4-benzylthioether complex having the highest number of total interactions (23) and hydrogen bond interactions (7 (Asp269, Ser214, Glu213, Glu144, Val251, Ala250, Hie267)) (Table 7 and Appendix A) could have contributed to epigallocatechin 4-benzylthioether having a significantly higher binding free energy over amoxicillin (Figure 13a), silicristin, propan-2-one, and chroman-4-one at the allosteric site of PBP2a (Appendix A). Nevertheless, this observation did not entirely justify PBP2a + epigallocatechin 4-benzylthioether having a significantly lower binding free energy relative to PBP2a + epicatechin gallate (Figure 13b) observed in this study. However, among the top-five phenolics, only epigallocatechin 4-benzylthioether failed to interact with some of the essential amino acids of the allosteric site (Table 7 and Appendix A). According to Alhadrami et al. [16], amino acids, such as Tyr297, Tyr105, Asn146, Ile144, and 296 are important at the allosteric site of PBP2a of *S. aureus*, all of which are absent in the interaction plot of PBP2a + epigallocatechin 4-benzylthioether, which could have impacted epigallocatechin 4-benzylthioether having a lower binding free energy relative to epicatechin gallate at the allosteric site. This observation is suggestive of the partial occupancy of epigallocatechin 4-benzylthioether at the allosteric site of PBP2a and hence its lesser potential to cause a higher fluctuation of Tyr446 (active site gatekeeper residue) relative to epicatechin gallate despite the high number of interactions. Moreover, epicatechin gallate at the allosteric site of PBP2a (11 interactions), interacted with a relatively shorter bond length (3.87 Å) when compared with epigallocatechin 4-benzylthioether and other studied compounds (Figure 13b and Table 7). In addition, despite having the least number of interactions among the top-five phenolics, PBP2a + epicatechin gallate alongside PBP2a + silicristin had the second-highest number of hydrogen bond interactions (five) (Figure 13b and Table 7). All these observations could have contributed to the significantly higher binding free energy noted in this study by epicatechin gallate at the allosteric site of PBP2a relative to epigallocatechin 4-benzylthioether and other study compounds and hence re-emphasises the impact of bond length and hydrogen bonds in the stability and affinity of a compound for a protein. PBP2a + silicristin, PBP2a + Propan-2-one, and PBP2a + Chroman-4-one all had interactions including hydrogen bonds that justified their respective binding free energy observed in this study (Table 7 and Appendix A). PBP2a + silicristin had sixteen interactions at the allosteric site of PBP2a, of which five interactions were hydrogen bond interactions with residues Lys273, Lys148, Asn146, and Asp295(2) and three interactions were other important interactions with Asp275, Lys316, and Tyr297. PBP2a + Propan-2-one had sixteen interactions at the allosteric site of PBP2a, of which four interactions were hydrogen bond interactions with residues Lys148, Lys273, Val277, and Ala276 and two interactions were other important interactions with Tyr297, Lys273. Similarly, PBP2a + Chroman-4-one had twelve interactions at the allosteric site of PBP2a, of which four interactions were hydrogen bond interactions with residues Lys148, Lys273, Val277, and Ala276 and one other important interaction with Tyr297. The extra hydrogen bond interaction and at least one other important interaction had by PBP2a + silicristin over PBP2a + Propan-2-one, and PBP2a + Chroman-4-one could have contributed to silicristin’s higher binding free energy over propan-2-one and chroman-4-one at the allosteric site of PBP2a. In this study, all the top-five phenolics interacted with a higher number of interactions, hydrogen bond interactions and other important interactions than amoxicillin (Figure 13a) at the allosteric site of PBP2a, which could have contributed to amoxicillin’s significantly lower binding free energy at the allosteric site of PBP2a relative to the top-five phenolics. PBP2a + amoxicillin had eight interactions at the allosteric site of PBP2a, of which two interactions were hydrogen bond interactions with residues Lys122 and Asp294 and two other important interactions with Lys122 and Glu212. Similar to PBP2a + epigallocatechin 4-benzylthioether, none of the interacted amino acid residues in the PBP2a + amoxicillin plot were among the essential amino acid residues at the allosteric site of PBP2a, suggesting the partial occupancy of this site by amoxicillin and could have imparted the lesser fluctuation of gatekeeper residue (Tyr446) observed in this study relative to the top-five phenolics and hence the lesser ability of amoxicillin to modulate the PBP2a active site when bound at the allosteric site.

## 4. Conclusions

Findings from this study have identified lead phenolics with the capability to bind to the active and allosteric sites of PBP2a of *S. aureus*. Specifically, the lead phenolics showed more promising affinity at the allosteric site than the active site, with silicristin and epicatechin gallate having the best affinity at the active and allosteric sites of PBP2a, respectively. Interestingly, the modulation of Tyr446, a residue implicated in impeding access of β-lactams to the PBP2a active site, was noted to correlate with the affinity of the lead phenolics with the allosteric site of PBP2a with epicatechin gallate, which has the highest binding free energy, resulting in the highest fluctuation of Tyr446. This observation is indicative of the lesser involvement of Tyr446 in the intramolecular interactions of adjacent residues of PBP2a, thereby creating lesser obstructions of the PBP2a active site. Furthermore, when the lead phenolics bind at the active site of PBP2a, Tyr446 had less fluctuation, implying its active involvement in the binding of the compounds. These observations are suggestive of the possible advantage of the lead phenolics in the direct and synergistic treatment of infections caused by *S. aureus* and, interestingly, as the lead phenolics had favourable pharmacokinetics and a feasible synthetic score, signifying the possibility of structural alterations for improved druggability and toxicity. However, further in vitro and in vivo confirmation of the activities observed in this study is strongly recommended and efforts are underway in this direction.

## Figures and Tables

**Figure 1 pharmaceutics-14-01818-f001:**
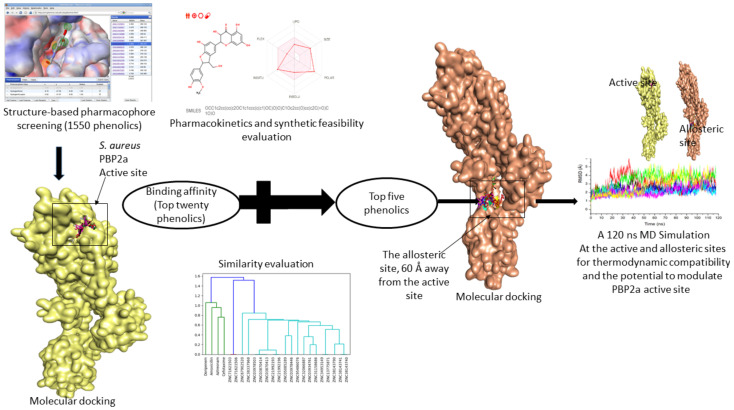
Workflow of the strategy adopted.

**Figure 2 pharmaceutics-14-01818-f002:**
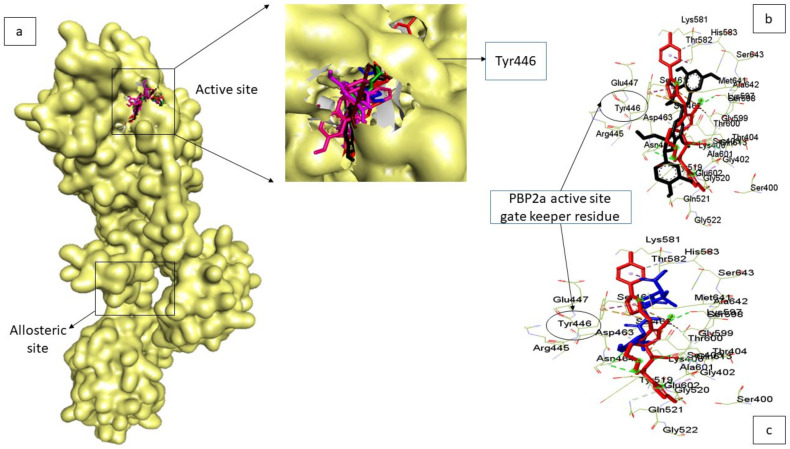
Validation of docking technique and parameters via the redocking approach against the co-crystal structure of PBP2a from *S. aureus* (3ZFZ). (**a**) the superimposition showed that the top-five phenolics (chroman-4-one (green), epicatechin gallate (purple), epigallocatechin 4-benzylthioether (brown), propan-2-one (pink), silicristin (black) and amoxicillin (blue)) could achieve the same orientation with the native inhibitor (red) of 3ZFZ with a low RMSD value of <1. (**b**,**c**) showed the superimposition of silicristin (phenolic with the highest docking score) and amoxicillin (antibiotic with the highest docking score) with the native inhibitor of 3ZFZ, displaying the amino acid at the active site (located 60 Å away from the allosteric site) and the active site gatekeeper residue (Tyr 446) [16].

**Figure 3 pharmaceutics-14-01818-f003:**
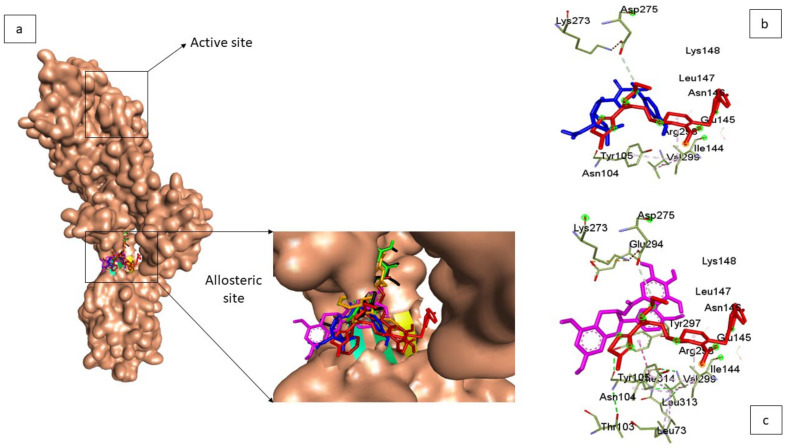
(**a**) Superimposition at the allosteric site of the co-crystal structure of PBP2a from *S. aureus* (3ZFZ), demonstrating the capability of the top-five phenolics (chroman-4-one (green), epicatechin gallate (purple), epigallocatechin 4-benzylthioether (brown), propan-2-one (pink), silicristin (black) and amoxicillin (blue)) to achieve the same orientation with the native inhibitor of 3ZFZ (red) with a low RMSD value of <1. (**b**,**c**) showed the superimposition of epicatechin gallate (phenolic with the highest docking score) and amoxicillin (antibiotic with the highest docking score) with the native inhibitor of 3ZFZ, displaying the amino acid at the allosteric site of PBP2a of *S. aureus* (located 60 Å away from the active site) [16].

**Figure 4 pharmaceutics-14-01818-f004:**
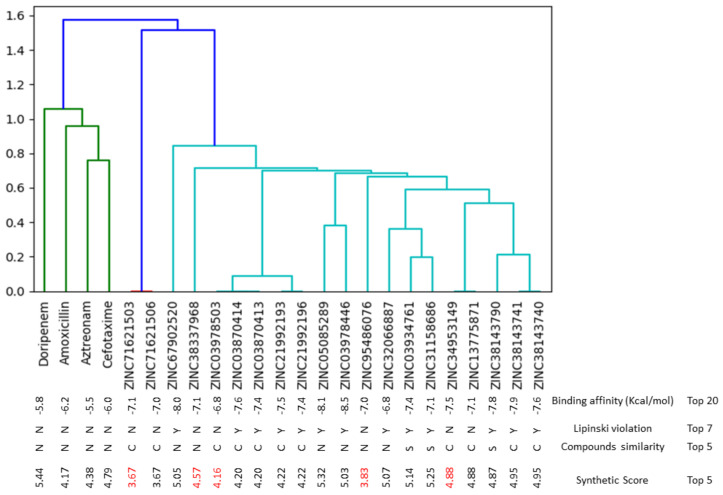
Molecular fingerprinting of the top-twenty phenolics. Compounds of the same colour and cluster were more similar than compounds of different colours and clusters. The top-twenty compounds were structurally different from the antibiotics which had the same colour (Green). Phenolics with similarity scores of zero and belonging to the same clusters were conformers, and in selecting the top five (highlighted in red colour), the conformer with the highest binding affinity that did not violate the Lipinski violation was selected among the top five. While avoiding the top-five phenolics that were conformers of each another, one common moiety that the top-five-ranked compounds had was resorcinol. Two (ZINC38337968 and ZINC03978503) of the top-five phenolics had a pyrogallol structure and from the cluster chart, these two compounds appeared to be the most similar among the top-five compounds. Only one (ZINC03978503) of the top-five phenolics had the catechol group. The top-five compounds all had a synthetic score of less than five.

**Figure 5 pharmaceutics-14-01818-f005:**
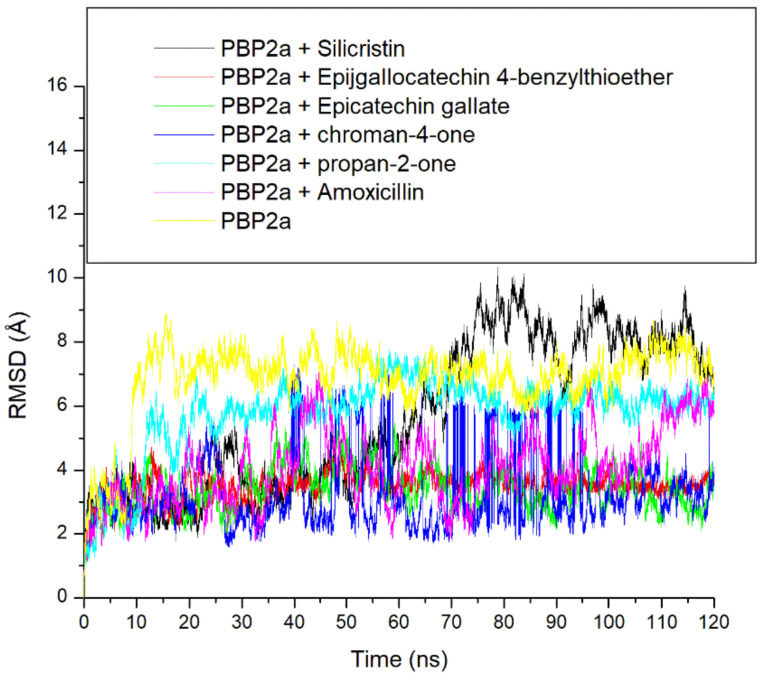
Comparative root-mean-squared deviation (RMSD) plots of alpha-carbon, the top-five phenolics, and amoxicillin against the active site of PBP2a of *S. aureus* over a 120 ns MD simulation period.

**Figure 6 pharmaceutics-14-01818-f006:**
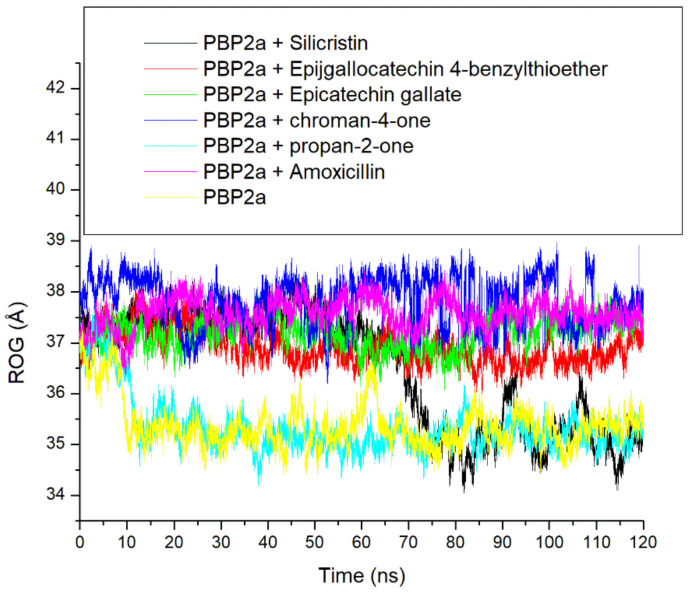
Comparative radius of gyration (ROG) plots of alpha-carbon, the top-five phenolics, and amoxicillin against the active site of PBP2a of *S. aureus* over a 120 ns MD simulation period.

**Figure 7 pharmaceutics-14-01818-f007:**
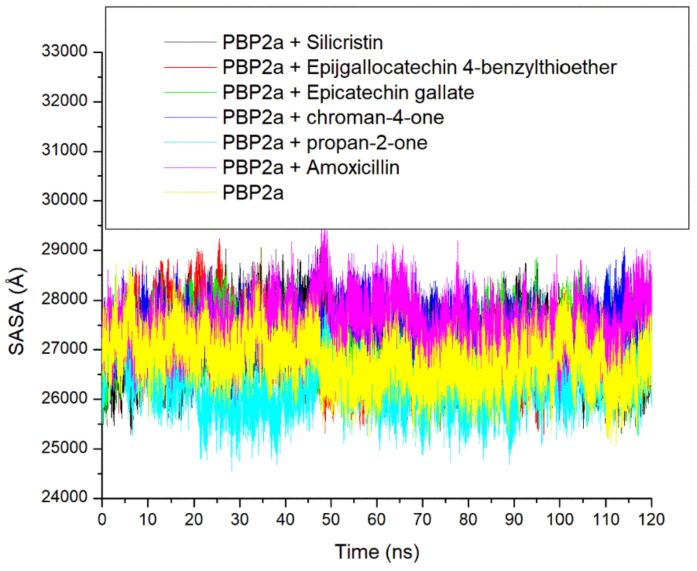
Comparative solvent-accessible surface area (SASA) plots of alpha-carbon, the top-five phenolics, and amoxicillin against the active site of PBP2a of *S. aureus* over a 120 ns MD simulation period.

**Figure 8 pharmaceutics-14-01818-f008:**
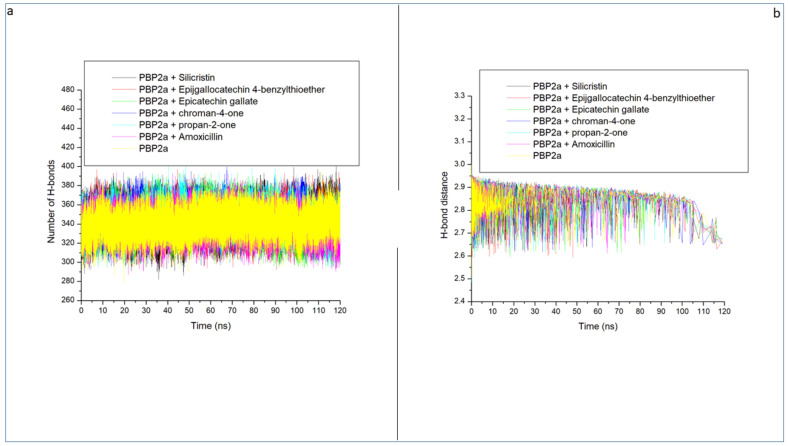
Time evolution of the number of intramolecular hydrogen bonds (**a**) and distance (**b**) formed in PBP2a following the binding of amoxicillin and the top-five phenolics at the active site of PBP2a of *S. aureus* during the 120 ns MD simulation period.

**Figure 9 pharmaceutics-14-01818-f009:**
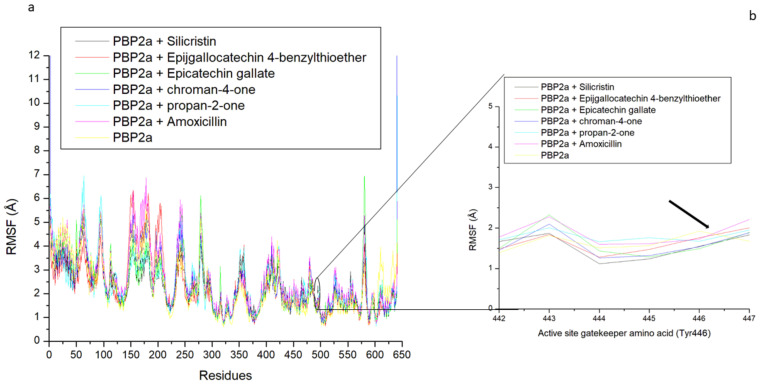
Comparative root-mean-square fluctuation (RMSF) plots of alpha-carbon, the top-five phenolics, and amoxicillin against residues of PBP2a of *S. aureus* (**a**), and gatekeeper residue [Tyr446 (black arrow)] of PBP2a (**b**) when the top-five phenolics and amoxicillin were bound at the active site of PBP2a over a 120 ns MD simulation period.

**Figure 10 pharmaceutics-14-01818-f010:**
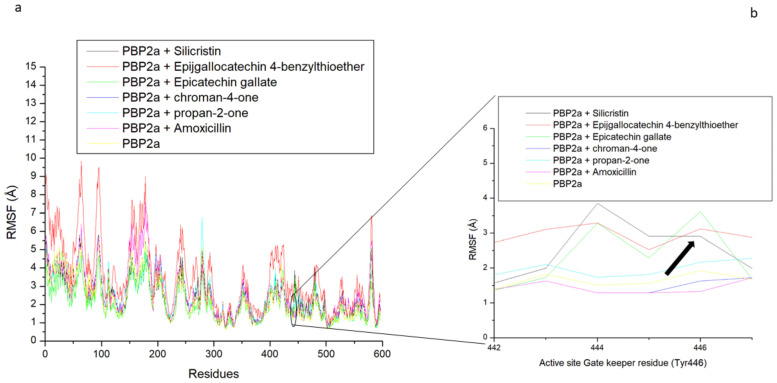
Comparative root-mean-square fluctuation (RMSF) plots of alpha-carbon, the top-five phenolics, and amoxicillin against residues of PBP2a of *S. aureus* (**a**), and gatekeeper residues [Tyr446 (black arrow)] of PBP2a (**b**) when the top-five phenolics and amoxicillin were bound at the allosteric site of PBP2a over a 120 ns MD simulation period.

**Figure 11 pharmaceutics-14-01818-f011:**
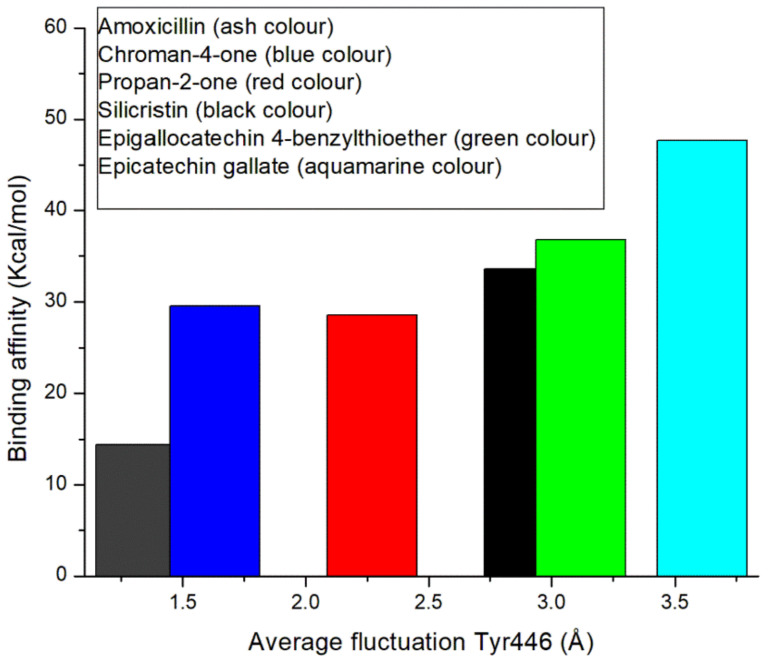
Correlation of the binding free energy of the top-five phenolics at the allosteric site of PBP2a and fluctuation of Tyr446 of PBP2a of *S. aureus*.

**Figure 12 pharmaceutics-14-01818-f012:**
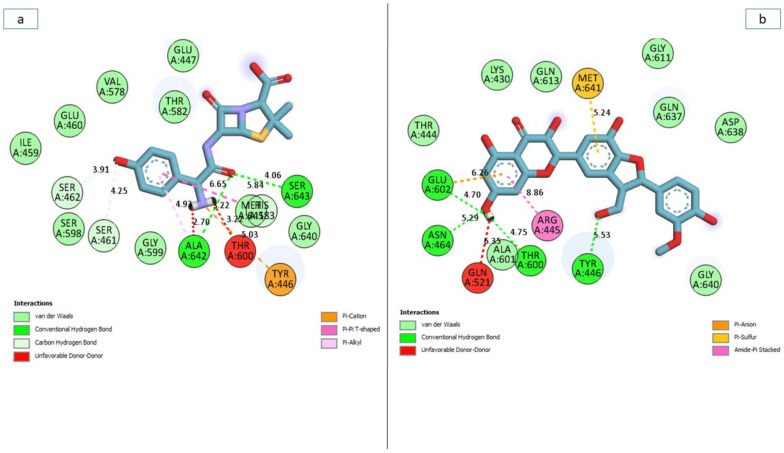
Plot of interactions of amoxicillin (**a**) and silicristin (**b**) against the active site of PBP2a of *S. aureus*.

**Figure 13 pharmaceutics-14-01818-f013:**
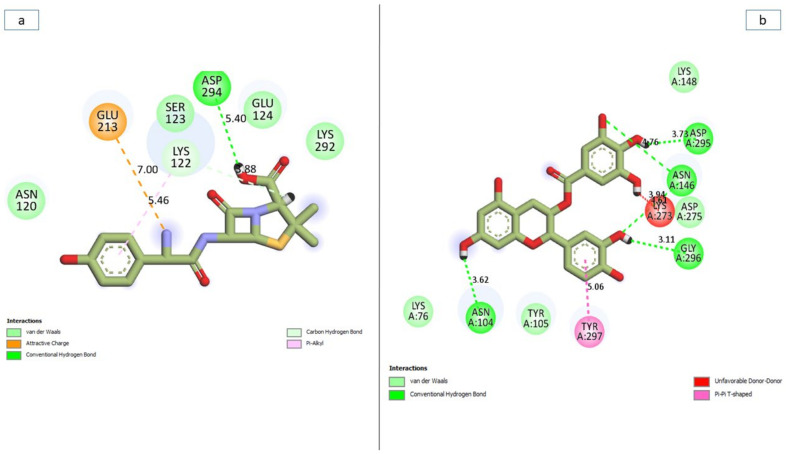
Plot of interactions of amoxicillin (**a**) and epicatechin gallate (**b**) against the allosteric site of PBP2a of *S. aureus*.

**Table 1 pharmaceutics-14-01818-t001:** Binding affinity of the top-five phenolics against the active site of PBP2a and their ADMET properties.

Ligands Zinc Code	Popular Name	B A (kcal/mol)	MW < 500 (g/mol)	HB-A ≤ 10	HB-D ≤ 5	Log *P*_o/w_ ≤ 5	RT-B ≤ 9	LV < 2	WS	GI -A	BS	pgp	Inhibitor of CYP 450s					H	C	IM	M	CY	TC	SA
													CYP 1A2	CYP 2C19	CYP 2C9	CYP 2D6	CYP 3A4							
Amoxicillin	Amoxicillin	−6.2	365.40	6	4	−0.39	5	N	VS	L	0.55	N	N	N	N	N	N	I	I	I	I	I	6	4.17
ZINC34953149	Silicristin	−7.5	482.44	10	6	1.49	4	N	S	L	0.55	N	N	N	N	N	Y	I	I	A	I	I	4	4.88
ZINC71621503	Propan-2-one	−7.1	458.46	7	6	3.3	6	N	MS	L	0.55	N	N	N	Y	N	Y	I	I	I	I	I	4	3.67
ZINC38337968	Epigallocatechin 4-benzylthioether	−7.1	428.46	7	6	2.23	4	N	MS	L	0.55	N	N	N	N	N	Y	I	I	I	I	I	5	4.57
ZINC95486076	Chroman-4-one	−7.0	356.37	6	3	3.09	3	N	MS	H	0.55	N	Y	N	Y	N	Y	I	I	A	I	I	4	3.83
ZINC03978503	Epicatechin gallate	−6.8	442.37	10	7	1.3	4	N	S	L	0.55	N	N	N	N	N	N	I	I	I	I	I	4	4.16

Keywords: BA: Binding affinity; MW: Molecular weight; HB-A: Hydrogen bond acceptor; HB-D: Hydrogen bond donor; Log P_o/w_: Partition coefficient; RT-B: rotatable bond; WS: Water solubility; GI- A: Gastrointestinal absorption; Pgp: Permeability glycoprotein substrate; CYP: Cytochrome; VS: very soluble; MS: Moderately soluble; S: Soluble; N: No; Y: Yes; L: low; I: Inactive; A: Active; LV: Lipinski violations; BS: Bioavailability score; H: Hepatotoxicity; C: Carcinogenicity; IM: Immunotoxicity; M: Mutagenicity; CY: Cytotoxicity; LD: Lethal dose; TC: Toxicity class; SA: Synthetic accessibility; and BA: binding affinity.

**Table 2 pharmaceutics-14-01818-t002:** Energy components (kcal/mol) of the top-five phenolics against the active site of PBP2a of *S. aureus*.

Systems	ΔE_vdW_	ΔE_elec_	ΔG_gas_	ΔG_solv_	ΔG_bind_
PBP2a + amoxicillin	−22.30 ± 7.99	−74.82 ± 26.08	−97.12 ± 28.39	75.58 ± 23.67	−21.54 ± 6.59
PBP2a + silicristin	−35.15 ± 8.08	−32.24 ± 12.41	−67.42 ± 15.90	41.80 ± 10.24	−25.61 ± 7.08
PBP2a + propan-2-one	−23.27 ± 5.46	−35.71 ± 16.25	−58.99 ± 18.62	39.92 ± 12.55	−19.06 ± 7.26
PBP2a + epigallocatechin 4-benzylthioether	−19.36 ± 4.64	−49.56 ± 13.09	−68.93 ± 11.38	44.18 ± 7.98	−24.75 ± 4.72
PBP2a + chroman-4-one	−22.23 ± 4.56	−47.43 ± 10.23	−65.56 ± 12.34	−42.78 ± 23.34	−22.34 ± 5.23
PBP2a + epicatechin gallate	−20.61 ± 4.45	−43.07 ± 16.16	−63.69 ± 16.86	40.58 ± 10.45	−23.11 ± 7.45

ΔE_vdW_ = van der Waals energy; ΔG_bind_ = total binding free energy; ΔE_gas_ = gas phase free energy; ΔE_elec_ = electrostatic energy; and ΔG_solv_ = solvation free energy.

**Table 3 pharmaceutics-14-01818-t003:** Average RMSD, ROG, RMSF, SASA, and intramolecular-hydrogen-bond number and distance values of the top-five phenolics following a 120 ns simulation at the active site of PBP2a of *S. aureus*.

Systems	RMSD (Å)	RMSF(Å)	ROG (Å)	SASA (Å)	Number of Intramolecular H-Bond	Intramolecular H-Bond Distance (Å)
Unbound PBP2a	6.86 ± 1.18	2.71 ± 0.91	35.37 ± 0.45	26786.05 ± 473.91	339.24 ± 12.45	2.85 ± 0.06
PBP2a + Amoxicillin	4.07 ± 1.15	2.85 ± 1.24	37.57 ± 0.29	27484.50 ± 458.31	341.44 ± 13.23	2.85 ± 0.06
PBP2a + Silicristin	5.65 ± 2.34	2.26 ± 2.09	36.47 ± 1.15	27160.02 ± 545.40	346.00 ± 12.15	2.85 ± 0.05
PBP2a + Propan-2-one	5.78 ± 1.21	2.45 ± 1.13	35.34 ± 0.55	26303.34 ± 464.78	342.94 ± 12.13	2.85 ± 0.06
PBP2a + Epigallocatechin 4-benzylthioether	3.49 ± 0.42	2.27 ± 1.14	36.92 ± 0.32	27060.31 ± 531.98	344.00 ± 12.22	2.85 ± 0.06
PBP2a + Chroman-4-one	3.26 ± 1.16	2.49 ± 3.22	37.84 ± 0.46	27311.74 ± 408.97	345.47 ± 12.13	2.85 ± 0.05
PBP2a + Epicatechin gallate	3.42 ± 0.67	2.38 ± 1.14	37.19 ± 0.29	27187.53 ± 411.42	343.16 ± 12.12	2.85 ± 0.06

**Table 4 pharmaceutics-14-01818-t004:** PBP2a active site gatekeeper residue average fluctuation (Å) after 120 ns simulation of the top-five phenolics at the active site.

PBP2a Active Site Gatekeeper Residue	Top-Five Phenolics						
	Silicristin	Epigallocatechin 4-benzylthioether	Epicatechin gallate	Chroman-4-one	Propan-2-one	Amoxicillin	Unbound PBP2a
Tyr446	1.55	1.76	1.49	1.53	1.67	1.72	1.92

**Table 5 pharmaceutics-14-01818-t005:** Molecular docking, binding free energy and RMSF values of the top-five phenolics at the allosteric site of PBP2a of *S. aureus*.

Systems	Binding Affinity (Kcal/mol)	ΔG_bind_ (Kcal/mol)	RMSF (Å)
Unbound PBP2a			2.71 ± 1.17
PBP2a + Amoxicillin	−7.7	−14.36 ± 6.00	2.56 ± 1.32
PBP2a + Silicristin	−8.4	−33.57 ± 5.38	2.45 ± 1.11
PBP2a + Propan-2-one	−8.3	−28.55 ± 8.07	2.49 ± 1.14
PBP2a + Epigallocatechin 4-benzylthioether	−8.1	−36.83 ± 8.04	3.48 ± 1.84
PBP2a + Chroman-4-one	−8.0	−29.52 ± 4.20	2.55 ± 1.32
PBP2a + Epicatechin gallate	−8.5	−47.65 ± 8.42	2.06 ± 0.95

ΔG_bind_ = total binding free energy.

**Table 6 pharmaceutics-14-01818-t006:** PBP2a active site gatekeeper residue average fluctuation (Å) after 120 ns simulation of the top-five phenolics at the allosteric site.

PBP2a Active Site Gatekeeper Residue	Top-Five Phenolics						
	Silicristin	Epigallocatechin 4-benzylthioether	Epicatechin gallate	Chroman-4-one	Propan-2-one	Amoxicillin	Unbound PBP2a
Tyr446	2.91	3.12	3.61	1.63	2.27	1.33	1.93

**Table 7 pharmaceutics-14-01818-t007:** Bond interactions against the active and allosteric site of PBP2a of *S. aureus* by the top-five phenolics.

Top-Five Phenolics and Standard	Total Number of Interactions (Average Distances in Å)	Number of Hydrogen Bonds and Interaction Residues	Other Important Interactions and Residues	Unfavourable Bonds (Bond Length in Å)
**Active Site**				
Amoxicillin	17 (5.14)	4 (Ala642, Ser643, Ser461, Ser462)	3 (Tyr446, Ala642, His583)	2 (Thyr600, Ala642) (2.96)
Silicristin	16 (5.75)	4 (Tyr446, Thyr600, Asn464, Glu602)	3 (Met641, Glu602, Arg445)	1 (Gln521) (6.35)
Propan-2-one	17 (5.53)	1 (Tyr447)	3 (His583, Tyr446, Ala642)	None
Epigallocatechin 4-benzylthioether	15 (4.91)	4 (Lys584, Glu460, Thr582, Asp586)	3 (Ala642, His583, Tyr446)	None
Chroman-4-one	15 (4.99)	1(Ser642)	2 (His583, Ala642)	None
Epicatechin gallate	15 (4.94)	5 (Lys581, Ser461, Glu447 (2), Gly599)	2 (Ala642, His583)	None
**Allosteric Site**				
Amoxicillin	8 (5.44)	2 (Lys122, Asp294)	2 (Lys122, Glu212)	None
Silicristin	16 (5.37)	5 (Lys273, Lys148, Asn146, Asp295(2))	3 (Asp275, Lys316, Tyr297)	(Tyr297) (6.06)
Propan-2-one	16 (5.23)	4 (Asp275, Lys273, Asn146 (2))	2 (Tyr297, Lys273)	none
Epigallocatechin 4-benzylthioether	23 (4.71)	7 (Asp269, Ser214, Glu213, Glu144, Val251, Ala250, Hie267)	2 (Val251, Pro332)	none
Chroman-4-one	12 (5.0)	4 (Lys148, Lys273, Val277, Ala276)	1 (Tyr297)	none
Epicatechin gallate	11 (3.87)	5 (Asn104, Gly296, Asn146 (2), Asp295)	1 (Tyr297)	Lys273 (4.61)

## Data Availability

The data are contained within the article or Appendix A.

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
