# Peer review of "Cheminformatics Identification of Phenolics as Modulators of Penicillin-Binding Protein 2a of Staphylococcus aureus: A Structure–Activity-Relationship-Based Study"

_pharmaceutics, 2022, doi:10.3390/pharmaceutics14091818_

Round 1
Reviewer 1 Report
Aribisala et al. submitted the manuscript, “Cheminformatic identification of phenolics as modulators of penicillin-binding protein 2a of Staphylococcus aureus: a structure-activity relationship-based study” is about naturally-derived phenolic structures, where authors used a virtual screening, molecular dynamics with the help of AutoDock tools, Chimera and other molecular modeling tools.
Please address these points.
1. Please maintain uniformity as there is an irregular use of Staphylococcus aureus and S. aureus.
2. Improve the choice of words.
Line 107, “The structure was cleaned” should be more scientific.
Line 146 “The docked complexes of the top five phenolics with the best orientation (maximum binding affinity)”
My suggestion is “best orientation” can be “most energy minimized”.
Add more information to this sentence.
3. How naturally-derived phenolics are suitable in particular to this target (PBP2a) and specify to S. aureus. Please elaborate and add more scientific information. Previous literature can be useful.
4. Validation of docking method and docking energy:
Molecular docking methods often produce pseudo-positive binding conformation as most energy-minimized pose. Therefore, docking studies generally require a parallel validation with cellular protein/enzyme affinity binding assays.
How did the authors confirm that their attained docking pose is not a pseudo-positive pose?
However, the authors could indeed argue that their MD simulation results are in agreement, but as MD simulation studies are generally performed after docking studies, therefore, MD simulations can’t measure the correctness of the docking methods in eliminating the pseudo-positive poses.
The authors also made some comparisons in the manuscript with respect to the docking energies as binding/affinity of particular phenolics towards the target. How did the authors evaluate the docking energy in the first place?
5. Authors rely on tools to evaluate the medicinal chemistry of their top phenolics: synthetic feasibility, drug-likeness, and toxicity.
These evaluations can provide some information regarding inhibitors pharmacokinetics in the early phase of drug discovery research but often produce pseudo-positive results.
What measures/tools did the authors choose/implement to reduce/eliminate such pseudo-positive results?
6. Formatting issues: Page numbers and formatting of the manuscript is improper. Please go through the manuscript.
The authors must revise the manuscript and address the comments mentioned above before getting considered for the journal.
Author Response
Dear Editor,
We appreciate you and the reviewers for your constructive comments. Below we provide a point-to-point response to the comments.
We hope you find these revisions relevant, and we are ready to make further amendments if deemed necessary.
Thank you.
Reviewer #1
Please address these points.
1. Please maintain uniformity as there is an irregular use of Staphylococcus aureus and S. aureus.
Response: Thank you for your observation; however, after careful proofreading of the manuscript, Staphylococcus aureus was only used at first mention in the introduction and/or at the beginning of a sentence, and subsequently, it was written as S. aureus.
- Improve the choice of words.
Line 107, "The structure was cleaned" should be more scientific.
Response: Thank you for the suggestion, the word “cleaned” has been changed to “prepared”
Line 146 "The docked complexes of the top five phenolics with the best orientation (maximum binding affinity)"
My suggestion is "best orientation" can be "most energy minimized".
Response: Thank you for the suggestion, the word "best orientation" has been changed to “most energy minimized conformation” as suggested
- How naturally-derived phenolics are suitable in particular to this target (PBP2a) and specify to S. aureus. Please elaborate and add more scientific information. Previous literature can be useful.
Response: In bacteria, the efficiency of β-lactam antibiotics has been attributed to their ability to penetrate the outer cell membrane, resist inactivation by plasmid or chromosomal encoded β-lactamase, and bind efficiently with mutated penicillin-binding proteins (e.g. PBP2a, a target that has been implicated in the resistant strain of S. aureus) (Zapun et al., 2008). These characteristics have been reported in phytonutrients, including phenolics (Zhao et al., 2002, Aldulaimi, 2017, Alhadrami et al., 2020). Phenolics' ability to effectively bind and modulate bacterial druggable targets such as β-lactamase and consequently inhibit various multi-drug resistant Gram-negative and Gram-positive bacteria has been reported (Aldulaimi, 2017, Aldulaimi et al., 2020), while also enhancing cell membrane permeability for antimicrobial absorption (Araya-Cloutier et al., 2018). Also, being the most frequently occurring metabolite in plants, phenolics offer other health and therapeutic benefits as antioxidants and against degenerative diseases such as cancer and diabetes (Weinreb et al., 2004; Panat et al., 2016; Dai et al., 2006, Aldulaimi, 2017, Aldulaimi et al., 2020). All of these characteristics of phenolics point to their prospective antibacterial capabilities to treat or act in synergy with conventional antibiotics in the treatment of infections caused by multi-drug resistant S. aureus. This information has been re-written more concisely in the introduction section of the manuscript.
References
Zapun et al. “Penicillin-binding proteins and β-lactam resistance” FEMS Microbiol. Rev., 32 (2008) 361–385. https://doi.org/10.1111/j.1574-6976.2007.00095.x
Zhao et al. “Inhibition of penicillinase by epigallocatechin- gallate resulting in restoration of antibacterial activity of penicillin against penicillinase- producing Staphylococcus aureus,” Antimicrob. Agents Chemother., 46 (2002) 2266-2268. https://doi.org/10.1128/AAC.46.7.2266-2268.2002
Panat et al. “Troxerutin, a natural flavonoid binds to DNA minor groove and enhances cancer cell killing in response to radiation” Chem. Biol. Interact., 5 (2016), 34-44. https://doi.org/10.1016/j.cbi.2016.03.024
Dai et al. “Fruit and vegetable juices and Alzheimer’s disease: The Kame project” Am J Clin Nutr., 119(2006) 751–759. https://doi.org/10.1016/j.amjmed.2006.03.045
Weinreb et al. “Neurological mechanisms of green tea polyphenols in Alzheimer’s and Parkinson’s diseases” J. Nutr. Biochem., 15 (2004) 506–516. https://doi.org/10.1016/j.jnutbio.2004.05.002
Alhadrami et al. “Flavonoids as potential anti-MRSA agents through modulation of PBP2a: A computational and experimental study” Antibiotics, 9 (2020) 562, https://doi.org/10.3390/antibiotics9090562
Aldulaimi, “General overview of phenolics from plant to laboratory, good antibacterials or not?,“Pharmacogn., 11 (2017) 123–127. https://doi.org/10.4103/phrev.phrev_43_16
Araya-Cloutier et al. “Rapid membrane permeabilization of Listeria monocytogenes and Escherichia coli induced by antibacterial prenylated phenolic compounds from legumes” Food Chem., 240 (2018) 147–155. doi: 10.1016/j.foodchem.2017.07.074
- Validation of docking method and docking energy:
Molecular docking methods often produce pseudo-positive binding conformation as most energy-minimized pose. Therefore, docking studies generally require a parallel validation with cellular protein/enzyme affinity binding assays.
How did the authors confirm that their attained docking pose is not a pseudo-positive pose?
However, the authors could indeed argue that their MD simulation results are in agreement, but as MD simulation studies are generally performed after docking studies, therefore, MD simulations can't measure the correctness of the docking methods in eliminating the pseudo-positive poses.
Response: Thank you for the question. The most common way to evaluate the correctness of a docking geometry is to measure the Root Mean Square Deviation (RMSD) of the ligand from its reference position in the answer complex after optimal superimposition (Kufareva et al., 2012). A low RMSD value of < 1 between the docked ligand from its reference position in the answer complex suggests the same binding orientation, which encouraged docking technique validation (Fahad et al., 2013). In this study, validation of the docking pose was done via the superimposition approach against the experimental co-crystal structure of PBP2a from S. aureus (3ZFZ). The superimposition showed that the top five phenolics and amoxicillin achieved the same orientation with the native inhibitor of 3ZFZ with a low RMSD value of < 1 which validates the docking scores observed in the study and this was presented in Figures 2 and 3 for PBP2a active and allosteric sites, respectively.
Reference
Fahad et al. “In Silico Prediction of Mechanism of Erysolin-induced Apoptosis in Human Breast Cancer Cell Lines” American, J. Bioinform., 3(2013) 62-71, DOI: 10.5923/j.bioinformatics.20130303.03
Kufareva and Abagyan, “Methods of protein structure comparison.” Methods Mol Biol. 2012; 857:231-57. doi: 10.1007/978-1-61779-588-6_10.
The authors also made some comparisons in the manuscript with respect to the docking energies as binding/affinity of particular phenolics towards the target. How did the authors evaluate the docking energy in the first place?
Response: The binding energies of the phenolics were evaluated using the binding affinity of standard beta-lactam antibiotics (amoxicillin, cefotaxime, aztreonam, and doripenem) commonly used against S. aureus as benchmarks. Following sorting, the top 20 phenolics were observed to have higher docking scores than all the reference standards. All of this information was included in the manuscript.
- Authors rely on tools to evaluate the medicinal chemistry of their top phenolics: synthetic feasibility, drug-likeness, and toxicity.
These evaluations can provide some information regarding inhibitors pharmacokinetics in the early phase of drug discovery research but often produce pseudo-positive results.
What measures/tools did the authors choose/implement to reduce/eliminate such pseudo-positive results?
Response: Thank you for the question. As rightly said, the evaluation of pharmacokinetics and medicinal chemistry of compounds via the in silico tools as used in this study sometimes can produce pseudo-positive results. However, to reduce false information, different online tools [SwissADME web and Molinspiration toolkits] equipped with models for robust prediction were employed in the study to enable comparison and validation of the pharmacokinetics and medicinal chemistry information of the phenolics. Moreover, further in vitro and in vivo validation of the prediction observed was strongly recommended and was clearly stated in the manuscript. I can also confirm that efforts are underway in our laboratory for both in vitro and in vivo validations.
6. Formatting issues: Page numbers and formatting of the manuscript is improper. Please go through the manuscript.
Response: Thank you for the observation. The page number issues and formatting errors in the manuscript have been resolved.
Reviewer 2 Report
The manuscript "Cheminformatic identification of phenolics as modulators of penicillin-binding protein 2a of Staphylococcus aureus: a structure-activity relationship-based study" was an interesting study. Authors explained and discussed the results precisely.
I recommend acceptance of the manuscript in its current form in journal 'Pharmaceutics'
Author Response
Reviewer #2
The manuscript "Cheminformatic identification of phenolics as modulators of penicillin-binding protein 2a of Staphylococcus aureus: a structure-activity relationship-based study" was an interesting study. Authors explained and discussed the results precisely.
I recommend acceptance of the manuscript in its current form in journal 'Pharmaceutics'
Response: Thank you for finding time to review the manuscript. We appreciate your kind comment.
Round 2
Reviewer 1 Report
The authors revised and answered the queries as asked, the manuscript can be accepted in the journal.